# ATP8B1 Knockdown Activated the Choline Metabolism Pathway and Induced High-Level Intracellular REDOX Homeostasis in Lung Squamous Cell Carcinoma

**DOI:** 10.3390/cancers14030835

**Published:** 2022-02-07

**Authors:** Xiao Zhang, Rui Zhang, Pengpeng Liu, Runjiao Zhang, Junya Ning, Yingnan Ye, Wenwen Yu, Jinpu Yu

**Affiliations:** 1Cancer Molecular Diagnostics Core, National Clinical Research Center of Caner, Key Laboratory of Cancer Prevention and Therapy, Key Laboratory of Cancer Immunology and Biotherapy, Tianjin Medical University Cancer Institute & Hospital, Tianjin 300060, China; zhangxiao123@tmu.edu.cn (X.Z.); zhangrui_zr@tmu.edu.cn (R.Z.); Liupengpeng123@tmu.edu.cn (P.L.); zhangrunjiao@tmu.edu.cn (R.Z.); ningjunya123@tmu.edu.cn (J.N.); yeyingnan123@tmu.edu.cn (Y.Y.); 2Tianjin’s Clinical Research Center for Cancer, Tianjin 300060, China; 3National Clinical Research Center of Caner, Tianjin’s Clinical Research Center for Cancer, Key Laboratory of Cancer Immunology and Biotherapy, Department of Immunology, Tianjin Medical University Cancer Institute & Hospital, Tianjin 300060, China; yuwenwne123@tmu.edu.cn

**Keywords:** ATP8B1, LUSC, CHKA, REDOX

## Abstract

**Simple Summary:**

We found that low expression of ATP8B1 was associated with poor prognosis, and involved in the dysregulation of glutathione (GSH) synthesis and choline metabolism in lung squamous cell carcinoma (LUSC) samples of The Cancer Genome Atlas (TCGA) and Tianjin Medical University Cancer Institute and Hospital (TJMUCH) cohort. We further constructed ATP8B1 knockdown of LUSC cell lines H520^SH^^-ATP8B1^ and SK-MES-1^SH^^-ATP8B1^ to investigate how ATP8B1 knockdown promoted cell proliferation, migration, and invasion in vitro and in vivo via upregulation of the CHKA-dependent choline metabolism pathway. We identified that ATP8B1 knockdown and CHKA upregulation can lead to mitochondrial dysfunction and high reduction-oxidation (REDOX) homeostasis, which may be involved in the roles of cardiolipin in maintaining mitochondrial dynamics and phospholipid homeostasis. Therefore, we proposed ATP8B1 as a novel predictive biomarker in LUSC and targeting ATP8B1-driven specific metabolic disorder might be a promising therapeutic strategy for LUSC.

**Abstract:**

The flippase ATPase class I type 8b member 1 (ATP8B1) is essential for maintaining the stability and polarity of the epithelial membrane and can translocate specific phospholipids from the outer membrane to the inner membrane of the cell. Although ATP8B1 plays important roles in cell functions, ATP8B1 has been poorly studied in tumors and its prognostic value in patients with lung squamous cell carcinoma (LUSC) remains unclear. By investigating the whole genomic expression profiles of LUSC samples from The Cancer Genome Atlas (TCGA) database and Tianjin Medical University Cancer Institute and Hospital (TJMUCH) cohort, we found that low expression of ATP8B1 was associated with poor prognosis of LUSC patients. The results from cellular experiments and a xenograft-bearing mice model indicated that ATP8B1 knockdown firstly induced mitochondrial dysfunction and promoted ROS production. Secondly, ATP8B1 knockdown promoted glutathione synthesis via upregulation of the CHKA-dependent choline metabolism pathway, therefore producing and maintaining high-level intracellular REDOX homeostasis to aggravate carcinogenesis and progression of LUSC. In summary, we proposed ATP8B1 as a novel predictive biomarker in LUSC and targeting ATP8B1-driven specific metabolic disorder might be a promising therapeutic strategy for LUSC.

## 1. Introduction

Lung cancer is a heterogeneous disease with extensive clinicopathological features, among which non-small cell lung cancer (NSCLC) accounts for 85% of the total diagnoses. Treatment of advanced lung cancer is based on the pathological subtype. Lung squamous cell carcinoma (LUSC) is the major subtype of NSCLC with limited treatment options [1], with low sensitivity to chemotherapy and a lack of targetable mutations and immune responses, which limits the benefits of chemotherapy, targeted therapy, and immune monotherapy [2,3]. Therefore, the discovery of predictive biomarkers and novel treatments for LUSC is urgently required. 

Metabolic reprogramming is a hallmark of cancer and in some cases, reprogrammed metabolic activities can be used to diagnose, monitor, and treat cancer. Tumor metabolism is a flexible network that enables tissues to meet the needs of homeostasis and growth and to evolve during tumor progression [4]. Lung cancer cells rewire their metabolic and energy production networks to support survival and rapid proliferation [5]. However, cancer metabolism is heterogeneous, so specific metabolic pathways should be targeted for intervention to achieve the best therapeutic effects [6].

Long interspersed nuclear element-1 (LINE-1) retrotransposition has become a major marker of cancer and has been considered as a cis-regulatory element to regulate the expression of host genes in recent years [7]. We previously discovered the role of LINE-1 in promoting the malignant progression of LUSC by regulating metabolism-related genes, in which LINE-1 interferes with the ATP8B1 gene and facilitates poor prognosis of LUSC [8]. 

ATP8B1 encodes a P4-ATPase, which is reported to be a flippase ATPase class I member and associated with progressive familial intrahepatic cholestasis (PFIC). ATP8B1 is essential for maintaining the stability and polarity of the epithelial membrane and can transport certain phospholipids (phosphatidylserine, cardiolipin) from the outer monolayer to the intima of the plasma membrane [9]. However, the function of ATP8B1 in carcinogenesis has been seldom studied [10,11], and downregulation of ATP8B1 was reported to be associated with poor prognosis in colorectal cancer without any further explanation of the underlying regulatory mechanism.

Here, we found that low expression of ATP8B1 was associated with poor prognosis and involved in the dysregulation of glutathione (GSH) synthesis and choline metabolism in LUSC samples of The Cancer Genome Atlas (TCGA) and Tianjin Medical University Cancer Institute and Hospital (TJMUCH) cohort. We further constructed ATP8B1 knockdown of LUSC cell lines H520^SH^^-ATP8B1^ and SK-MES-1^SH^^-ATP8B1^ to investigate how ATP8B1 knockdown promoted cell proliferation, migration, and invasion in vitro and in vivo via upregulation of the CHKA-dependent choline metabolism pathway. We identified that ATP8B1 knockdown and CHKA upregulation can lead to mitochondrial dysfunction and high reduction-oxidation (REDOX) homeostasis, which may be involved in the roles of cardiolipin in maintaining mitochondrial dynamics and phospholipid homeostasis [12]. Therefore, we proposed ATP8B1 as a novel predictive biomarker in LUSC and targeting ATP8B1-driven specific metabolic disorder might be a promising therapeutic strategy for LUSC.

## 2. Results

### 2.1. Low Expression of ATP8B1 Was Associated with Poor Prognosis of LUSC

According to our previous research [8], we analyzed the frequency of 13 LINE-1 insertions in 109 sequenced tissue samples from the TJMUCH cohort, and found that the occurrence frequency of L1-ATP8B1 was the highest in LUSC patients suffering from a shorter survival (survival time ≤ 40 months) (Figure 1A). The patients were stratified into two groups by the expression level of L1-ATP8B1 by RT-qPCR and the transcriptome sequencing data indicated that ATP8B1 mRNA expression was downregulated in the L1-ATP8B1^+^ group (Figure 1B), suggesting low expression of ATP8B1 might play a role in promoting LUSC progression. According to the transcriptome results, we then divided all samples into two groups according to the expression level of the ATP8B1 gene: the ATP8B1 low expression (ATP8B1^low^) group and ATP8B1 high expression (ATP8B1^high^) group. A high ratio of L1-ATP8B1^+^ samples was observed in ATP8B1^low^ tumor tissues (Figure 1B). The survival analysis using the Kaplan–Meier method revealed that low expression of ATP8B1 was significantly correlated with poor prognosis in the TJMUCH cohort (Figure 1C). Further, we obtained clinical information from 407 LUSC samples from The Cancer Genome Atlas (TCGA) dataset, and divided them into two groups according to the expression level of ATP8B1. The result of the survival analysis in the TCGA dataset was consistent with that in the TJMUCH cohort (Figure 1D). We compared differentially expressed genes between the ATP8B1^low^ and ATP8B1^high^ groups either in the TJMUCH cohort or TCGA database, followed by pathway enrichment analysis based on Kyoto Encyclopedia of Genes and Genomes (KEGG). Most differentially expressed genes were enriched in metabolic pathways, including drug metabolism, serotonergic synapse, retinol metabolism, ascorbate and aldarate metabolism, metabolism of xenobiotics by cytochrome P450, and glutamatergic synapse (Figure 1E,F). These results implied that dysregulation of metabolic pathways might be involved in low expression of ATP8B1 associated with poor prognosis of LUSC.

### 2.2. ATP8B1 Knockdown Promoted Proliferation, Inhibited Apoptosis, and Aggravated Invasion and Migration of LUSC Cells In Vitro and In Vivo

To further investigate the roles of ATP8B1 in cell functions, we infected LUSC cell lines (H520 and SK-MES-1) with a recombinant lentivirus with the ATP8B1 shRNA sequence to knockdown the ATP8B1 gene (H520^SH^^-ATP8B1^ and SK-MES-1^SH^^-ATP8B1^). Using H520 cells without any treatment for control normalization, we compared the expression of ATP8B1 in H520 cells infected with control lentivirus or SH-ATP8B1 lentivirus, and found that the mRNA level of ATP8B1 was significantly reduced in H520^SH^^-ATP8B1^(Figure 2A). Consistent results were detected in SK-MES-1^SH^^-ATP8B1^. Using the Cell Counting Kit 8 (CCK8) proliferation assay, we found that the proliferation rate of either H520^SH^^-ATP8B1^ or SK-MES-1^SH^^-ATP8B1^ significantly increased compared with the controls (Figure 2B). Moreover, the results of the MTT assay showed that the cell viability increased in H520^SH^^-ATP8B1^ and SK-MES-1^SH^^-ATP8B1^ (Figure 2C). The Annexin-V apoptosis assay showed that the apoptotic rates in H520^SH^^-ATP8B1^ and SK-MES-1^SH^^-ATP8B1^ decreased (Figure 2D). We then evaluated the roles of ATP8B1 in cell migration and invasion via the wound healing and trans-well invasion assay. The results showed that the wound closure rates (WCRs) of H520^SH^^-ATP8B1^ and SK-MES-1^SH^^-ATP8B1^ were significantly higher than those of the controls (Figure 2E). Similarly, the cell numbers of H520^SH^^-ATP8B1^ and SK-MES-1^SH^^-ATP8B1^ that migrated through the matrigel layer after 48h were significantly higher, indicating that the invasion ability was significantly enhanced by knockdown of ATP8B1 (Figure 2F). Furthermore, H520^SH^^-ATP8B1^, SK-MES-1^SH^^-ATP8B1^, and the control cells were subcutaneously injected into NOD-SCID mice. After 24 days, the average volume of xenografts in the H520^SH^^-ATP8B1^ and SK-MES-1^SH^^-ATP8B1^ groups was at least 2-fold higher than that in the control group. Accordingly, the tumor growth rate was significantly higher in the H520^SH^^-ATP8B1^ and SK-MES-1^SH^^-ATP8B1^ groups compared to that in the control group (Figure 2G). In conclusion, ATP8B1 knockdown promoted proliferation, inhibited apoptosis, and aggravated invasion and migration in vitro and in vivo.

### 2.3. ATP8B1 Knockdown Activated the Choline Metabolic Pathway via Upregulation of Choline Kinase Expression

We performed metabolomics analysis on H520^SH^^-ATP8B1^ and compared it with H520^SH^^-NC^. The most highly differentially activated metabolic pathway was the choline metabolic pathway in H520^SH^^-ATP8B1^ (Figure 3A,B), in which the levels of choline decreased and phosphorylcholine increased (Figure 3C). Since choline is phosphorylated by choline kinase α (CHKA) to form phosphorylcholine, which is an important precursor of phosphatidylcholine and necessary for cell growth [13], we detected the expression of CHKA by Western blot. The result revealed that the level of CHKA protein was higher in H520^SH^^-ATP8B1^ compared to controls, which was consistent with SK-MES-1^SH^^-ATP8B1^ (Figure 3D). Among the primary LUSC tissues from the TJMUCH cohort, the mRNA and protein levels of CHKA were upregulated in the ATP8B1^low^ group (Figure 3E,F). Similarly, the results of the tissue immunofluorescence assay identified that the expression level of CHKA was negatively correlated with that of ATP8B1 in LUSC cells, with a mutually exclusive pattern (Figure 3G). As phospholipases (including PLA, PLC, and PLD) are essential mediators in the catabolic process of phosphatidylcholine in the choline metabolism pathway [14,15,16], we detected the expression levels of PLA, PLC, and PLD in H520^SH^^-ATP8B1^ cells by Western blot. We found that the protein expression levels of PLC and PLD did not change significantly while only PLA was upregulated in H520^SH^^-ATP8B1^ cells (Appendix A), which can decompose phosphatidylcholine into fatty acids, thus promoting fatty acid metabolism and providing energy for cell growth [16]. These results indicated that ATP8B1 knockdown activated the choline metabolic pathway via upregulation of CHKA expression, thereby increasing cellular phosphorylcholine.

### 2.4. ATP8B1 Knockdown Promoted Cell Proliferation and Invasion in LUSC in A CHKA-Dependent Manner

In order to further investigate the roles of CHKA in ATP8B1-related carcinogenesis and tumor progression, we then interfered with the CHKA gene via siRNAs in H520^SH^^-ATP8B1^ cells. The mRNA and protein levels of CHKA were dramatically reduced in H520^SH^^-ATP8B1^^-si^^-CHKA^ cells (Figure 4A,B). Besides that, the levels of metabolites involved in the choline metabolism pathway, phosphorylcholine and phosphatidylcholine, were both decreased significantly in cells with CHKA knockdown (Figure 4C). We found that after CHKA knockdown, the proliferation rate of H520^SH^^-ATP8B1^ cells significantly decreased (Figure 4D), and the apoptotic rate accordingly increased (Figure 4E). Furthermore, after CHKA knockdown, the migration and invasion capacity of H520^SH^^-ATP8B1^ cells significantly decreased (Figure 4F,G). As an oncogene, CHKA is highly expressed in a variety of tumors. It can has a carcinogenic role by activating the MAPK and PI3K/AKT signaling pathways [17]. We found that the phosphorylation level of ERK and AKT was increased in H520^SH^^-^^ATP8B1^ cells. When we knocked down CHKA, the phosphorylation level of ERK and AKT was significantly reduced (Figure 4H). All the results in SK-MES-1 cells were consistent with those in H520 cells (Appendix A). These results further suggest that ATP8B1 knockdown promoted cell proliferation and invasion in LUSC in a CHKA-dependent manner.

### 2.5. ATP8B1 Knockdown Activated the Choline Metabolism Pathway and Induced High-Level Intracellular REDOX Homeostasis in LUSC

The mechanism by which CHKA activation plays a tumor-promoting role in ATP8B1 knockdown cell lines remains unclear. KEGG and GO enrichment analysis of highly differentially expressed genes in H520^SH^^-ATP8B1^ cells found that they were mainly enriched in cellular metabolic pathways, which indicated the correlation between ATP8B1 knockdown and upregulation of intracellular reduction-oxidation (REDOX) homeostasis, including the oxidation-reduction process, cell REDOX homeostasis, reactive oxygen species (ROS) metabolic process, and regulation of cell growth pathways (Figure 5A,B). Since glutathione (GSH) plays a role in inducing and maintaining REDOX homeostasis in cancer cells [18], we examined the level of GSH in H520^SH^^-ATP8B1^ cells. The GSH level was higher in H520^SH^^-ATP8B1^ cells compared to the controls while the ROS level was higher in H520^SH^^-ATP8B1^ cells compared to the controls (Figure 5C,D). After CHKA knockdown, the level of GSH decreased but the level of ROS increased significantly in H520^SH^^-ATP8B1^ cells (Figure 5D). We disrupted the intracellular REDOX homeostasis by inhibiting GSH synthesis and stimulating ROS production via buthionine sulfoximine (BSO) [19], and found that the intracellular levels of GSH deceased and ROS increased in H520^SH^^-ATP8B1^ cells simultaneously (Figure 5C,D). In addition, we found that BSO significantly inhibited cell proliferation, migration, and invasion but stimulated cell apoptosis of H520^SH^^-ATP8B1^ cells (Figure 5E–H). When we treated cells with BSO, we also found that the phosphorylation level of ERK and AKT decreased (Figure 5I). These results imply that ATP8B1 knockdown activated the choline metabolism pathway and induced high-level intracellular REDOX homeostasis to promote carcinogenesis and progression in LUSC. 

## 3. Discussion

ATP8B1 has mostly been studied in progressive familial intrahepatic cholestasis type 1 (PFIC1) [20]. It has only been demonstrated that its low expression was involved in the development of colorectal cancer through regulation of phospholipid homeostasis [10]. We reported for the first time that ATP8B1 regulates choline metabolism to maintain REDOX homeostasis and promotes carcinogenesis and progression of LUSC.

Activated choline metabolism is a hallmark of cancer [21,22], which leads to elevated phosphorylcholine levels in multiple types of cancer, triggering a more aggressive cancer phenotype [16,23,24]. Through gene expressing profiling analysis of LUSC tissue samples from the TCGA database and TJMUCH cohort, we found that low expression of ATP8B1 was associated with poor prognosis, and involved in dysregulation of choline metabolism. Metabolomics analysis indicated that ATP8B1 knockdown activated the choline metabolic pathway via upregulation of CHKA expression, therefore decreasing the level of intracellular choline and increasing the level of phosphorylcholine. 

CHKA has an enzymatic activity that phosphorylates choline to form phosphorylcholine. High expression of CHKA is associated with a higher tumor grade and poor prognosis of NSCLC [25], and silencing of CHKA reduces cell proliferation, migration, and invasion capability [26]. In this study, after CHKA knockdown, we observed that the cell proliferation, invasion, and migration capacity of H520^SH^^-ATP8B1^ and SK-MES-1^SH^^-ATP8B1^ cells was significantly reduced, which indicates that ATP8B1 knockdown promoted carcinogenesis and progression of LUSC in a CHKA-dependent manner. Moreover, CHKA is usually considered as an oncogene that activates the MAPK and PI3K/AKT pathways [17]. We found that the phosphorylation level of ERK and AKT increased in H520^SH^^-ATP8B1^ and SK-MES-1^SH^^-ATP8B1^ cell lines. When CHKA was knocked down, the phosphorylation level decreased. Therefore, blocking the choline metabolic pathway via a CHKA inhibitor might be a promising therapeutic strategy for LUSC. Until now, 14 CHKA inhibitors have been proposed at Research Collaboratory for Structural Bioinformatics (RCSB), among which TCD-717 has completed a phase I clinical trial and displayed long-term toxic antitumor efficacy in colon adenocarcinomas, non-lung squamous cell carcinomas, and breast adenocarcinomas [27,28,29,30].

Cancer cells can extensively reprogram their metabolic pathways to fulfill increased biosynthesis and bioenergy needs. The metabolism reprogramming of cancer cells always increases oxidative stress [18]. It was reported that ATP8B1 encodes a cardiolipin transporter that mediates the flip of cardiolipin from the outside to the inside of the mitochondrial inner membrane [4]. Since cardiolipin was reported to interact directly with a number of essential protein complexes on the mitochondrial membrane, including respiratory chain complexes I-V [5], ATP8B1 deficiency might lead to decreased mitochondrial complex enzyme activity via cardiolipin remodeling, resulting in ROS production upregulation [12]. Transcriptome analysis showed that most of the differentially expressed genes were enriched in the REDOX homeostasis pathway in ATP8B1-knockdown LUSC cells. REDOX homeostasis refers to the balance between oxidizing and reducing metabolites in cells. Compared with normal tissues, ROS production increases in tumor tissues, and the expression of antioxidant proteins, which are both involved in maintaining intracellular REDOX homeostasis to promote tumor cell signaling and maintain resistance to apoptosis [31].

In this study, we noticed that the glutamate energy metabolic pathway, which can regulate GSH metabolism and maintain the REDOX balance in cells, was highly differentially expressed either in LUSC tissues with low expression of ATP8B1 or LUSC cells with ATP8B1 knockdown [32,33,34]. It has been reported that CHKA silencing can significantly inhibit the content of GSH, thereby impairing antioxidant cell defense and possibly enhancing cellular response to drug therapy [23]. We proved that interference in CHKA expression in ATP8B1 knockdown cell lines could reduce the GSH content and increase the ROS level. Furthermore, the GSH inhibitor BSO could dramatically disrupt the high-level intracellular REDOX homeostasis to inhibit cell proliferation, invasion, and migration of LUSC cells as efficiently as CHKA knockdown. The destruction of REDOX homeostasis can also inhibit the activation of the MAPK and PI3K/AKT signaling pathway [35,36]. Here, we validated that interference with CHKA and REDOX homeostasis can decrease the phosphorylation level of both ERK and AKT, suggesting that ERK and AKT activation could be regulated by REDOX homeostasis (Appendix A).

In conclusion, ATP8B1 knockdown leads to abnormal phospholipid metabolism both on the cell membrane and mitochondrial membrane. The abnormal phospholipid metabolism on the mitochondrial membrane might cause dysfunction of the electron transport chain in mitochondria, resulting in increased ROS. In contrast, the abnormal phospholipid metabolism on the cell membrane might participate in choline metabolism reprogramming, CHKA activation and GSH production, and even oxidative stress resistance, which facilitate the maintenance of high-level intracellular REDOX homeostasis in cells, thus activating the MAPK and PI3K/AKT pathway to promote carcinogenesis and progression of LUSC cells. Therefore, for the first time, we propose ATP8B1 as a novel predictive biomarker in LUSC and targeting ATP8B1-driven specific metabolic disorder might be a promising therapeutic strategy for LUSC.

## 4. Materials and Methods

### 4.1. Patient Information

We obtained 109 cases of partial lung resection performed in Tianjin Medical University Cancer Institute and Hospital (TJMUCH) from the Cancer Biobank of TJMUCH (Table 1 and Table 2). This project was approved by the Ethics Committee of Tianjin Medical University (Approved No.: Ek2021143) and written informed consent was obtained from the patients. All experiments were performed in accordance with the principles of the Declaration of Helsinki.

### 4.2. Cell Lines and Cell Treatment

NCI-H520 and SK-MES-1 were purchased from Cellcook Co., Ltd. (Guangzhou, China) with cell authentication via the STR multi-amplification method. NCI-H520 was cultured in RPMI1640 (Gibco BRL). SK-MES-1 was cultured in Eagle’s Minimum Essential Medium. All medium contained 10% FBS and 1% penicillin/streptomycin. The cells were cultured at 37°, under 5% CO_2_. BSO (yuanye, Shanghai, China) was dissolved in H_2_O. BSO was diluted to 249.7uM, followed by replacing the cell medium 12h after cells seeded. All experiments were completed less than 2 months after establishing stable cell lines or thawing early passage cells.

### 4.3. Mouse Models

NOD-SCID mice, 5 weeks old, weighing about 17–18 g, were obtained from Beijing Vital River Laboratory Animal Technology Co., Ltd. (Beijing, China). All mice are housed in the SPF animal facility. We obtained different H520 and SK-MES-1 cells carrying constructed lentiviruses (SH-NC and SH-ATP8B1) and resuspended 5 × 104 cells in 100 μL of PBS with Matrigel, followed by administration of the cells to the flanks of NOD-SCID mice via subcutaneous injection. Each group included 5 mice. After the mice were constructed, the weight and tumor size of each mouse were monitored every 4 days. The calculation formula of the tumor volume (V) is V = π × L × W × H/6 (L: length, W: width, H: height). All animal protocols were approved by the Ethics Committee for Animal Experiments of TJMUCH (Approved No.: NSFC-AE-2021179), and were performed in accordance with the Guide for the Care and Use of Laboratory Animals. The Wistar IACUC guideline was followed to determine the time at which the survival experiments were ended (tumor burden exceeds 10% of the body weight).

### 4.4. Lentivirus Construction and Plasmid Transfection

For SH-ATP8B1 insertion lentivirus construction, the SH-ATP8B1 fragment was amplified by PCR using the complementary DNA (cDNA) of H520 and SK-MES-1 cells as a template. Then, the amplified SH-ATP8B1 fragment was inserted into Phblv-CMV-MCS-EF1-T2A-Puro lentiviral vectors (Hanbio Co., Ltd., Shanghai, China) and the constructed positive plasmid was confirmed by DNA sequencing.

### 4.5. RNA Extraction, Reverse Transcription-Polymerase Chain Reaction (RT-PCR), and Quantitative Real-Time PCR (qPCR) Analysis

As mentioned earlier, RNA extraction, cDNA synthesis, RT-PCR, and qPCR analysis were performed in accordance with the manufacturer’s protocol. The primer sequence is shown in Appendix A.

### 4.6. RNA Library Preparation, Sequencing, and Enrichment Analysis

In short, the library was sequenced on Illumina^®^ (NEB, Ipswich, MA, USA) according to the manufacturer’s recommendations. The RNA sequencing data were uploaded to the GEO database (accession number: GSE181042).

### 4.7. Immunohistochemistry

All procedures are executed as described above. The antibodies we used here are shown in Appendix A and biotinylated secondary goat anti-mouse IgG antibody (Santa Cruz, CA, USA), according to the manufacturer’s instructions, using the DAB staining kit (Maixin Biotechnology, Beijing, China) with streptavidin-horseradish peroxidase (HRP) labeling. For the negative control, IgG1 was used to replace each primary antibody. Positive staining areas were counted in 5 fields at 200× magnification.

### 4.8. Multispectral Fluorescent Immunohistochemistry

First, the slide was heated, then xylene was used to remove residual paraffin, and then rehydrated in graded alcohol. AR6 buffer and microwave processing were used for antigen retrieval. Then, the blocking solution was used for blocking. Then, the first primary antibody, position1, was applied and allowed to incubate. The opal polymer HRP Ms + Rb (Perkin Elmer, Hopkington, MA, USA) was used as the secondary antibody. The slides were washed and Tyramine Signal Amplification (TSA) dye was applied at position 1 (Opal 7 Color Kit, Perkin Elmer, Hopkington, MA, USA). The slides were then microwaved to peel off the primary and secondary antibodies, washed, and blocked again with a blocking solution. The second primary antibody was applied at position 2, and the process repeated, in which DAPI was applied after the unbound DAPI was washed away, and the slide was covered with LongtTM Gold anti-quenching reagent (Invitrogen, Carlsbad, CA, USA). Five fields at 200× magnification from the single-color slides were imaged, and StrataQuest Image Analysis software (v.6.0.1.181) was used to generate a spectral library for unmixing.

### 4.9. Cell Proliferation Assay

The Cell Counting Kit 8 (CCK8, Beyotime Biotechnology, Shanghai, China) assay was used to detect the proliferation ability. Cells were inoculated in 96-well plates at a density of 2 × 10^3^ cells/100 μL/well. Then, 10 μL of CCK-8 reagent were added to each well. The cells were incubated at 37 °C for 2 h and the absorbance at 450/650 nm was measured by a microplate reader (Synergy HT). The mean values of the three replicates were calculated.

The cells were inoculated into 96-well plates at a density of 3000 cells/well and grown in 1640/MEM supplemented with 10% FBS. Cell viability was determined by Solarbio’s MTT reagent (Solarbio, Beijing, China). On the day of determination, 10 μL of MTT solution were added and incubated at 37 °C for 4 h in fresh 1640/MEM + 10% FBS medium. Another 110 μL of Formazan solution were added. The light absorption at 490 nm was measured using microplate reader (Synergy HT) quantification. Each experiment was repeated at least three times.

### 4.10. Cell Apoptosis Assay

An Annexin-V-APC Apoptosis Assay Kit (BioBay, Tianjin, China) was used to detect cell apoptosis. After the cells were collected, they were washed with PBS and resuspended at a concentration of 1 × 10^5^ cells/mL to prepare a single-cell suspension. Subsequently, 5 μL of Annexin-V and 5 μL of APC were added to the cell suspension and the mixture was incubated in the dark for 15 min. The proportion of apoptotic cells (Annexin-V+ APC-) was determined by flow cytometry.

### 4.11. Wound Healing Assay

Cells were inoculated in 6-well plates. When the density reached 90~100%, a line was drawn on the monolayer cells with a 20 μL pipette tip to form “scratches”. The cells were rinsed with PBS to remove cell debris, and then cultured in fresh medium. In total, 5 fields were selected under the microscope at 0 and 48 h after scribbling. The scratch distance was measured three times each time to find the average value. Cell migration rate = (0 h scratch distance–48 h scratch distance)/0 h scratch distance × 100%.

### 4.12. Trans-Well Invasion Assay

The invasion ability was detected using Matrigel gel and the Trans-Well plate. The cells were inoculated in Matrigel and 200 μL of serum-free RPMI-1640 at a density of 2 × 10^4^ cells. The cells were inoculated in a 24-well plate trans-well system with an 8 μm polycarbonate filter membrane in the lower chamber and with medium containing 10% FBS. After incubation for 48 h, cells on the submembrane surface were fixed with methanol and stained with crystal violet. The stained membrane was photographed with a microscope. The number of cells was counted from five randomly selected fields, and the average value of the three complex pores was calculated.

### 4.13. Western Blot

Protein was quantified using the BCA protein assay kit (Solarbio, Beijing, China). Protein (20μg) was separated by 10% sodium dodecyl sulfate polyacrylamide gel electrophoresis and transferred to a polyvinylidene fluoride membrane (PVDF). The membrane was blocked with 5% skim milk, and then incubated with the primary antibody overnight at 4 °C. The antibodies used are shown in Appendix A. After incubating with HRP-conjugated α-rabbit or α-mouse secondary antibody for 1 h, the ChemiDoc imaging system (Bio-Rad, Hercules, CA, USA) was used to detect protein bands with a chemiluminescent substrate (Perkin Elmer).

### 4.14. Metabonomics

Metabolomics analysis was performed by Shanghai Bioprofile Technology Company Ltd. Cells were washed three times with sterile pre-cooled PBS and harvested for metabolic analysis. Then, a 100 mg sample was incorporated into 1 mL of cold methanol/acetonitrile/H_2_O (2:2:1, *v*/*v*/*v*) and adequately vortexed. The lysate was homogenized by an MP homogenizer. The homogenate was sonicated at a low temperature (30 min/once, twice). The mixture was centrifuged for 20 min. The supernatants were collected and dried in a vacuum centrifuge. Then, the lyophilized samples were reconstituted in 100 μL of acetonitrile/water (1:1, *v*/*v*) and adequately vortexed, and then centrifuged. The supernatants were collected for HPLC-MS/MS analysis using an UHPLC (1260 Infinity LC, Agilent Technologies, Santa Clara, CA, USA) coupled to a triple quadrupole mass spectrometer (Agilent 6460 QqQAgilent Technologies).

### 4.15. Determination of Intracellular Reactive Oxygen Species (ROS)

ROS was determined using a ROS detection kit (Beyotime Biotechnology, Shanghai, China) according to the manufacturer’s instructions. The cells were collected and incubated with DCFH-DA at 37 °C for 30 min, and then washed with serum-free medium. Flow cytometry was used to detect the DCF fluorescence of 1 × 10^7^/mL cells at an excitation wavelength of 488 nm and emission wavelength of 525 nm.

### 4.16. Glutathione (GSH) Detection Assay

The level of GSH was measured in cell or tumor lysates according to the instructions of the GSH and GSSG Assay Kit (Beyotime Biotechnology, Shanghai, China).

### 4.17. Phosphorylcholine Detection

We used an ELISA kit (Jianglaibio, Shanghai, China) to detect intracellular phosphorylcholine and phosphatidylcholine.

### 4.18. Statistical Analysis

Follow the manufacturer’s instructions, we analyzed the data using SPSS 22.0 and GraphPad Prism 8.0 software. The measurement data are expressed as the median (interquartile range) and were compared using the χ^2^ test. Quantitative data are presented in the form of average ± standard deviation, and were compared through analysis of variance and the LSD test. The Spearman rank test and linear regression analysis were used to evaluate the correlation between the expression levels detected by RT-qPCR. Univariate Cox regression and multivariate Cox regression analysis were used to identify common genes related to OS, and the cumulative survival rate was determined by the Kaplan–Meier method. The univariate survival rate of LUSC patients with different LINE-1 insertion and OS was analyzed by the two-sided log-rank test. The statistical significance was set as *p* < 0.05.

## 5. Conclusions

In conclusion, for the first time, we demonstrated that ATP8B1 could be used a novel predictive biomarker in LUSC. Our important findings of the oncogenic role of ATP8B1 knockdown-driven metabolic disorder in LUSC carcinogenesis are likely applicable to other cancers. We propose that targeting ATP8B1-driven specific metabolic disorder might be a promising therapeutic strategy for LUSC.

## Figures and Tables

**Figure 1 cancers-14-00835-f001:**
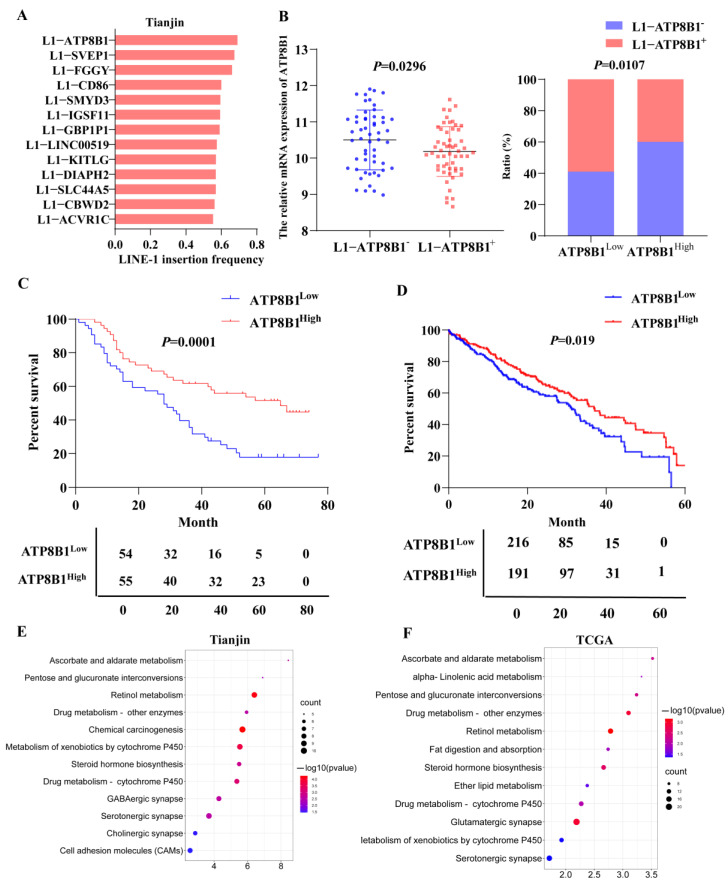
Low expression of ATP8B1 promotes poor prognosis for the tumor by regulating metabolism. (**A**) The frequency of 13 LINE-1 insertions in LUSC tissue samples from the TJMUCH cohort. The occurrence frequency of L1-ATP8B1 was the highest in the group with poor prognosis (survival time ≤ 40). (**B**) We divided LUSC samples from the TJMUCH cohort into L1-ATP8B1^+^ and L1-ATP8B1^−^ groups according to the relative mRNA expression level of L1-ATP8B1. The transcriptome sequencing data indicated that ATP8B1 mRNA expression was downregulated in the L1-ATP8B1^+^ group. In 109 LUSC tissues, ATP8B1 expression was divided into high- and low-expression groups according to the median of the expression values, and a high ratio of L1-ATP8B1^+^ samples was observed in ATP8B1^low^ tumor tissues. (**C**) Low expression of ATP8B1 was significantly associated with poor prognosis (*p* = 0.0001). (**D**) Low expression of ATP8B1 was associated with poor prognosis in patients with LUSC at 5-year survival in the TCGA database (*p* = 0.041). (**E**) Pathway enrichment of differentially expressed genes was associated with ATP8B1 in 109 tissues. (**F**) Pathway enrichment of differentially expressed genes was associated with ATP8B1 in the TCGA database.

**Figure 2 cancers-14-00835-f002:**
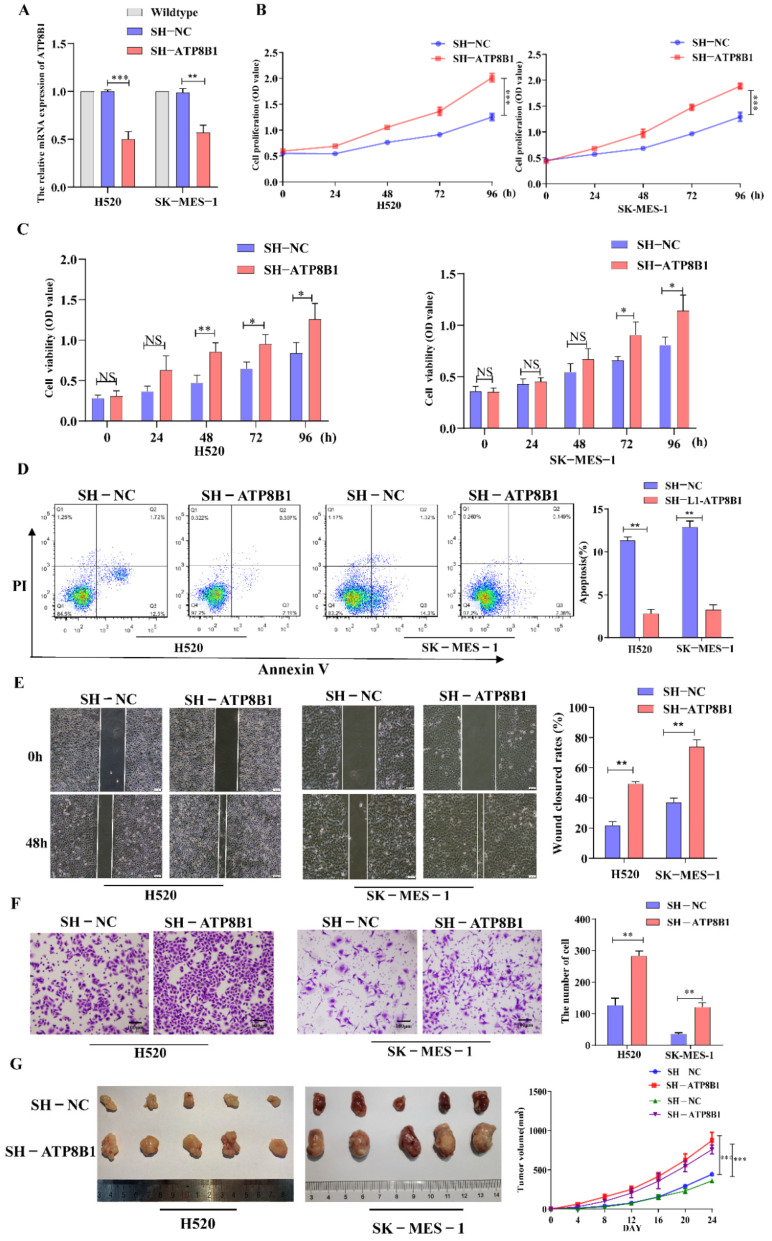
Interference of ATP8B1 promotes malignant progression in vitro and in vivo. (**A**) Expression of ATP8B1 in H520 and SK-MES-1 cells. All the RT-qPCR results shown for H520^SH^^-NC^ and H520^SH^^-ATP8B1^ were relative mRNA expression values compared to wildtype H520 cell without virus infection for normalization while all the RT-qPCR results shown for SK-MES-1^SH^^-NC^ and SK-MES-1^SH^^-ATP8B1^ were relative mRNA expression values compared to wildtype SK-MES-1 cells. (**B**) Cell proliferation results. (**C**) MTT assay results. (**D**) Cell apoptosis results. (**E**) Representative images of the wound healing assays. (**F**) Representative images of the trans-well invasion assays. (**G**) H520^SH^^-ATP8B1^, SK-MES-1^SH^^-ATP8B1^, and their control cells were subcutaneously injected into NOD-SCID mice as xenografts. Representative images of the forming tumors and the tumor volume growth curve at various time points are shown. * *p* < 0.05; ** *p* < 0.01; *** *p* < 0.001; Scale bar: 100 μm.

**Figure 3 cancers-14-00835-f003:**
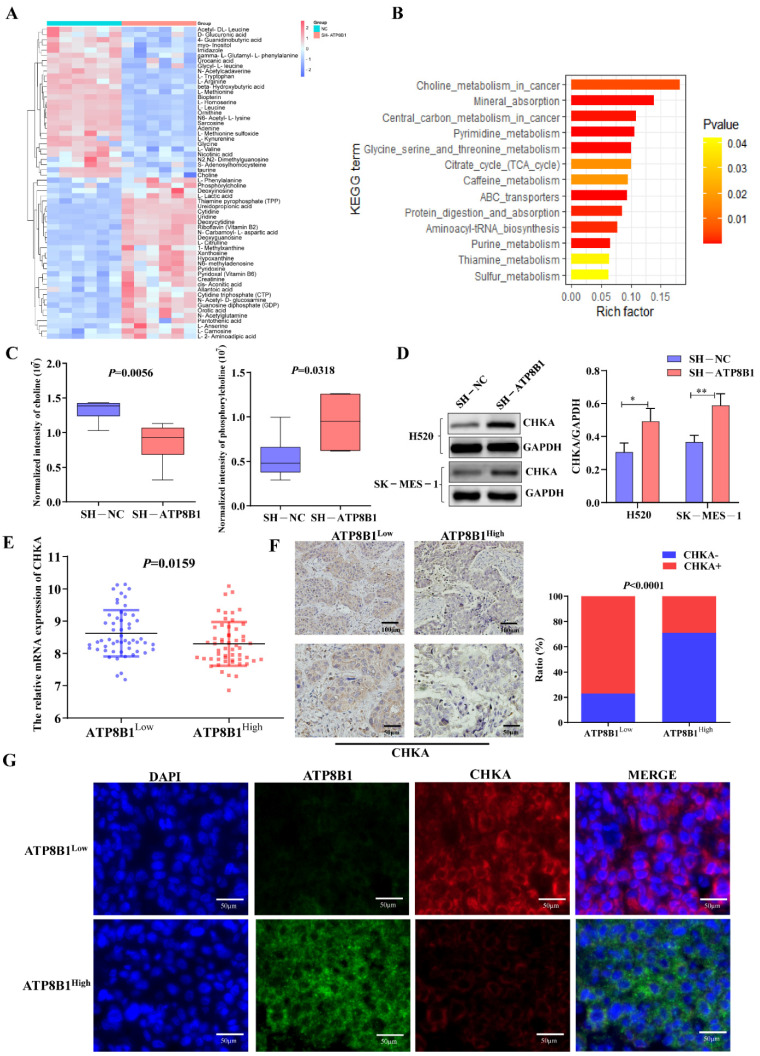
Low expression of ATP8B1 activates choline kinase and promotes the production of phosphorylcholine. (**A**) Six pairs of H520^SH^^-ATP8B1^ and H520^SH^^-NC^ samples were tested using metabolomics analysis. (**B**) KEGG pathway enrichment was carried out for different metabolites. (**C**) The cell metabolomics results showed that choline levels decreased and phosphorylcholine levels increased in the ATP8B1 knockdown cell lines. (**D**) Upregulated CHKA protein expression was detected in H520^SH^^-ATP8B1^ at the protein level in both cell lines. Original blots see Appendix A. (**E**) The results showed the relative mRNA expression of CHKA from the transcriptome sequencing data of TJMUCH LUSC tissues. (**F**) IHC results in 46 cases of LUSC showed that CHKA was highly expressed in the ATP8B1^low^ group. (**G**) Multispectral fluorescent immunohistochemistry proved that in LUSC, CHKA was highly expressed in the ATP8B1^low^ group. * *p* < 0.05; ** *p* < 0.01. Scale bar: 50 μm; 100 μm.

**Figure 4 cancers-14-00835-f004:**
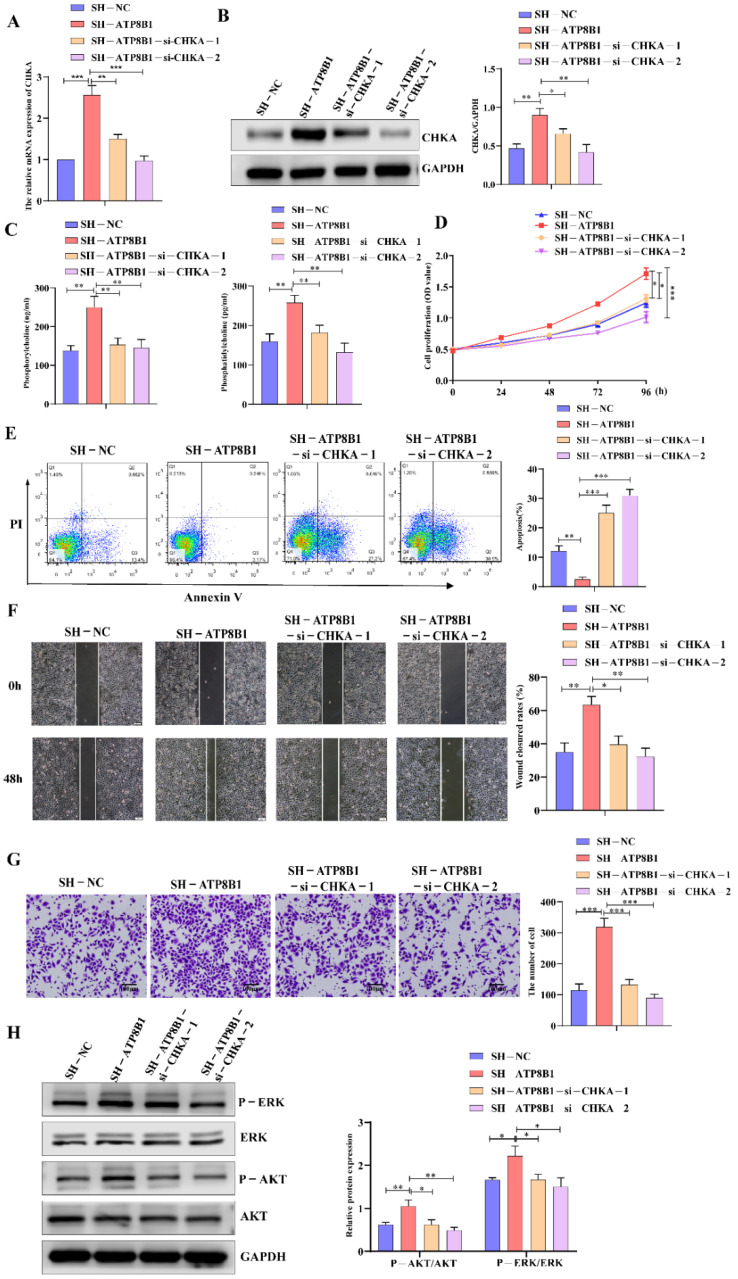
Interfering with CHKA expression in H520^SH^^-ATP8B1^ inhibited tumor progression. (**A**) Expression of CHKA in different cells at the mRNA level. All the RT-qPCR results shown for H520^SH^^-ATP8B1^ and H520^SH^^-ATP8B1-si^^-CHKAsi^^-CHKA^ were relative mRNA expression values compared to H520^SH^^-NC^ for normalization. (**B**) Expression of CHKA in different cells at the protein level. (**C**) The ELISA results of phosphorylcholine and phosphatidylcholine. (**D**) Cell proliferation results. (**E**) Cell apoptosis results. (**F**) Representative images of the wound healing assays. (**G**) Representative images of the trans-well invasion assays. (**H**) Analysis of phosphorylated ERK and phosphorylated AKT by Western blot. Original blots see Appendix A. * *p* < 0.05; ** *p* < 0.01; *** *p* < 0.001; Scale bar: 100 μm.

**Figure 5 cancers-14-00835-f005:**
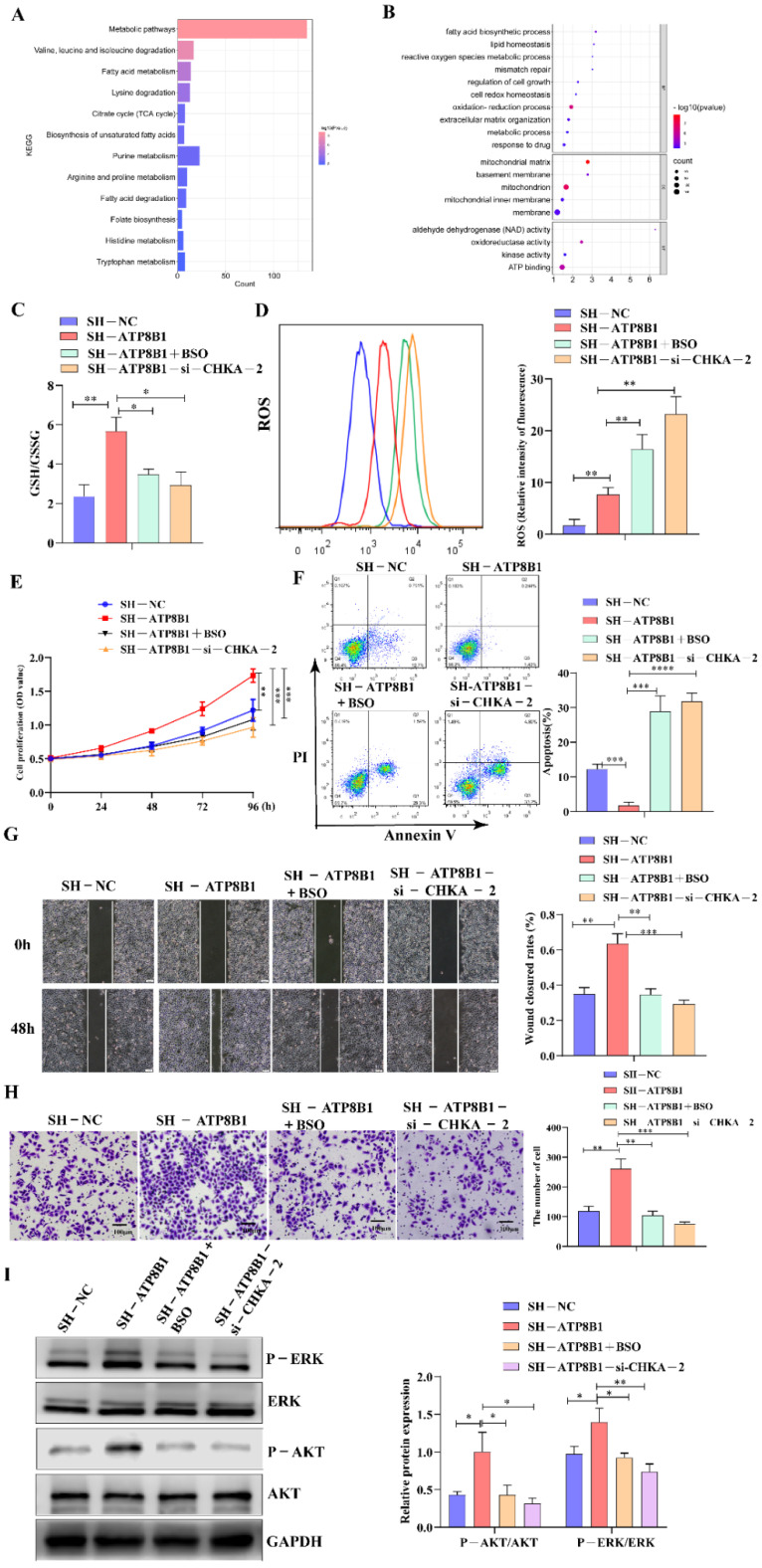
CHKA maintains REDOX balance by regulating the level of intracellular GSH. (**A**) KEGG pathway enrichment analysis of differential genes in H520^SH^^-ATP8B1^ and H520^SH^^-NC^ by transcriptome sequencing. (**B**) GO pathway enrichment analysis of differential genes in H520^SH^^-ATP8B1^ and H520^SH^^-NC^. (**C**) The level of intracellular GSH/GSSG. (**D**) Flow cytometry results of intracellular ROS. (**E**) Cell proliferation results. (**F**) Cell apoptosis results. (**G**) Representative images of the wound healing assays. (**H**) Representative images of the trans-well invasion assays. (**I**) Analysis of phosphorylated ERK and AKT by Western blot. Original blots see Appendix A. * *p* < 0.05; ** *p* < 0.01; *** *p* < 0.001; **** *p* < 0.0001; Scale bar: 100 μm.

**Table 1 cancers-14-00835-t001:** The basic clinical pathological information of all patients.

Clinical Pathological Parameters	Number of Patients
Total	109
Gender	
Male	81
Female	28
Age	
<60	52
≥60	57
Stage	
I–II	67
III–IV	42
T stage	
1–2	76
3–4	33
N stage	
0–1	77
2–3	32
M stage	
0	96
1	13
Metastatic site	
Negative	96
Lung	6
Bone	5
Brain	1
Others	2
Location	
Central	50
Periphery	59
Smoking	
Negative	20
Positive	89

**Table 2 cancers-14-00835-t002:** Distributions of the clinical pathological parameters of patients with different ATP8B1 expression levels.

Clinical Pathological Parameters	ATP8B1		*p*
	Low Expression	High Expression	
No. of patients	54	55	-
Gender			0.094
Male	40	48	
Female	14	7	
Age			0.567
<60	26	30	
≥60	28	25	
OS			0.004
≤40	38	23	
>40	16	32	
T stage			0.002
1–2	30	46	
3–4	24	9	
N stage			0.003
0–1	31	23	
2–3	23	9	
M stage			0.042
0	44	52	
1	10	3	
Location			0.848
Periphery	30	29	
Central	24	26	
Smoking			0.332
Negative	12	8	
Positive	42	47	

## Data Availability

The data presented in this study are all available within the article or Appendix A.

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
