# Peer review of "ATP8B1 Knockdown Activated the Choline Metabolism Pathway and Induced High-Level Intracellular REDOX Homeostasis in Lung Squamous Cell Carcinoma"

_cancers, 2022, doi:10.3390/cancers14030835_

Round 1
Reviewer 1 Report
Dear Editor,
The authors of the above manuscript highlight the implication of the flippase ATPase class I type 8b member 1 (ATP8B1) as a prognostic marker in lung squamous cell carcinoma (LUSC). Through in vitro and vivo experiments, they have found that ATP8B1 knockdown promoted proliferation, migration and invasion of LUSC cells via upregulation of choline kinase α (CHKA)-dependent pathway which is implicated in high-level of intracellular ROS production and promoted LUSC carcinogenesis and progression. Altogether, the main purpose of the manuscript is to provide new insights on the possibility to target the ATP8B1 driven specific metabolic response to develop new therapeutic strategies for LUSC. However, I would like to suggest some revisions before acceptance for publication.
- Figure 1A and B are to me quite confusing. Proper labelling of the axis should be reported and, in the text, should be clearly described the subgroups of L1-ATP8B1+/– and how they have been characterized also in terms of ATP8B1 expression. Also, the transcriptome results by which samples are divided in high and low ATP8B1 expression should be reported in the paper maybe in a Supplementary section.
- Figure 2A miss the labelling with control normalization used for the RT-qPCR. Authors used apoptosis and cell proliferation assays to test the impact of ATP8B1 KO in cells, however I would suggest to perform also an MTT assay as test of viability as this test is a measure of mitochondria activity, which in the contest of the study could be relevant.
- Figure 3A: the metabolomic analysis is missing in the paper nor described in the method section and should be included
Figure 3C: through metabolomic analysis the author found that choline metabolism is altered in H520-ATP8B1 KO cells and found increased CHKA expression. However, PC-PLC enzyme is also important for phosphatidylcholine (PC) catabolic pathway and has also implication in different types of cancer growth (Spadaro et al. Cancer Res 2008; Abalsamo et al Breast Cancer Research 2010 ) and is might be useful to test the expression levels and eventually the contribution of this enzyme in the ATP8B1 low cells/tissues. Furthermore, since CHKA has been proposed to be an oncogene (Ramirez et al. JBC 2008) as it sustains tumor growth via activation of Ras and PI3K signaling, authors should check if silencing of CHKA in ATP8B1-si (or using small CHKA antagonists like CK37, RSM-932A) affect those signaling pathways.
In Figure 3C I would correct the labeling of the graph.
Figure 3E: relative expression towards a control is missing in the labeling
Figure 3F-G: I would optimize the resolution of the images and especially for Figure G for which I have noticed a difference in the brightness of the pictures among single channels and the merged ones. Every picture has to be taken with the same intensity I would add also a quantification for that and a more detailed description in the method section.
- Figure 4A: a proper labeling for mRNA expression is missing
Figure 4B GAPDH is missing in the labeling
Figure 4G: labeling of the quantification should be consistent throughout the paper.
In addition, I would add more information in the Figure legends
- Figure 6: the final figure needs some improvements, it lacks of labeling (COX is missing) and information both in the figure and in the figure legend and I would suggest to use a bigger font for ATP8B1 and explain the correlation with cardiolipin.
- Check for typos and flow in the main text. Improve the method section and Figure legends as they lack informations.
Author Response
Dear Reviewer,
Thank you for your letter stating your consideration of a revision of the manuscript entitled “ATP8B1 knockdown activated choline metabolism pathway and induced high-level intracellular REDOX homeostasis in lung squamous cell carcinoma”.
As indicated in your letter, we would like to resubmit our revised manuscript to Cancers for consideration of publication. My co-authors and I would like to thank you for their thoughtful comments and thorough review of our manuscript with the intent to strengthen our manuscript further. In acknowledgment of your comments, we are very pleased to recognize that you appreciate the potential significance and impact of our work. Therefore, in accordance with your comments we have added significant new data, described in detail below, and revised the manuscript to address your concerns. Below, we provide the following responses in a rebuttal to each comment.
- Figure 1A and B are to me quite confusing. Proper labelling of the axis should be reported and, in the text, should be clearly described the subgroups of L1-ATP8B1+/– and how they have been characterized also in terms of ATP8B1 expression. Also, the transcriptome results by which samples are divided in high and low ATP8B1 expression should be reported in the paper maybe in a Supplementary section.
We are grateful for the reviewer’s constructive suggestion and comment. Firstly, as the reviewer suggested to present the data more clearly, we focused on the frequency of 13 LINE-1 insertions in LUSC tissue samples from TJMUCH cohort in Figure 1A, in which the occurrence frequency of L1-ATP8B1 was the highest in LUSC patients suffering from shorter survival (OS≤40 months). And we added the description of “LINE-1 insertion frequency” for X axis in Figure 1A to make the labeling more proper. Secondly, in Figure 1B, in order to describe the subgroups of L1-ATP8B1+/– were characterized in terms of ATP8B1 expression, we added the data of dividing LUSC samples according to the expression level of ATP8B1 gene in the transcriptome analysis. A high ratio of L1-ATP8B1+ samples was observed in ATP8B1 low expression (ATP8B1low) group compared to that in ATP8B1 high expression (ATP8B1high) group. As the reviewer suggested, we revised the description of “The relative mRNA expression of ATP8B1” for Y axis in Figure 1B to present the results more clearly and precisely. Additionally, we added the description for subgroup definition of either L1-ATP8B1+/– or ATP8B1high/low in detail in the text and figure legend (Figure 1A-B, line 115).
- Figure 2A miss the labelling with control normalization used for the RT-qPCR. Authors used apoptosis and cell proliferation assays to test the impact of ATP8B1 KO in cells, however I would suggest to perform also an MTT assay as test of viability as this test is a measure of mitochondria activity, which in the contest of the study could be relevant.
We appreciated the reviewer’s constructive suggestion and comment. We revised the labeling of Figure 2A as “The relative mRNA expression of ATP8B1”. And all the RT-qPCR results shown for H520SH-NC and H520SH-ATP8B1 were relative mRNA expression values compared to H520 cells without any treatment for control normalization, while all the RT-qPCR results shown for SK-MES-1SH-NC and SK-MES-1SH-ATP8B1 were relative mRNA expression values compared to SK-MES-1 cells without any treatment for control normalization. We supplemented the description in detail in the revised manuscript (Figure 2A, line 156). Furthermore, as suggested by the reviewer, we performed MTT assay to test cell viability. The result is consistent with the cell proliferation results using CCK-8 assay and was added into Figure2C and the Result section (Figure 2C, line 156).
- Figure 3A: the metabolomic analysis is missing in the paper nor described in the method section and should be included.
We appreciated the reviewer’s constructive suggestion and comment. We have supplemented the description of metabolomics analysis in the section of Materials and Methods (line 436-line 447).
- Figure 3C: through metabolomic analysis the author found that choline metabolism is altered in H520-ATP8B1 KO cells and found increased CHKA expression. However, PC-PLC enzyme is also important for phosphatidylcholine (PC) catabolic pathway and has also implication in different types of cancer growth (Spadaro et al. Cancer Res 2008; Abalsamo et al Breast Cancer Research 2010 ) and is might be useful to test the expression levels and eventually the contribution of this enzyme in the ATP8B1 low cells/tissues. Furthermore, since CHKA has been proposed to be an oncogene (Ramirez et al. JBC 2008) as it sustains tumor growth via activation of Ras and PI3K signaling, authors should check if silencing of CHKA in ATP8B1-si (or using small CHKA antagonists like CK37, RSM-932A) affect those signaling pathways.
We appreciate the reviewer’s constructive suggestion and comment. As phospholipases (including PLA, PLC and PLD) are essential mediators in the catabolic process of phosphatidylcholine in the choline metabolism pathway, we detected the expression levels of PLA, PLC and PLD in H520SH-ATP8B1 cells by western blot. We found that the protein expression levels of PLC and PLD did not change significantly in H520SH-ATP8B1 cells, while only PLA was up-regulated in H520SH-ATP8B1 cells, which can decompose phosphatidylcholine into fatty acids, thus promoting fatty acid metabolism and providing energy for cell growth. We have added this part of result in Figure S1 and the Result section (Figure S1A-B, line 596).
Since CHKA has been proposed to be an oncogene via activation of Ras and PI3K signaling, especially by increasing phosphorylated ERK and phosphorylated AKT, we detected the phosphorylation levels of ERK and AKT in H520SH-ATP8B1 and H520SH-ATP8B1-si-CHKA cells. We found that the levels of phosphorylated ERK (p-ERK) and phosphorylated AKT (p-AKT) were upregulated in H520SH-ATP8B1 cells, while downregulated after CHKA knockdown (Figure 4H, line 216). Furthermore, we added BSO (GSH inhibitor) to disrupt the cellular REDOX homeostasis in H520SH-ATP8B1 cells, and found that the levels of p-ERK and p-AKT were decreased significantly (Figure 5I, line 247). The results suggested that ATP8B1 knockdown promotes REDOX homeostasis through activation of CHKA, and then activates MAPK and PI3K/AKT pathways, thereby promoting tumor proliferation, invasion and migration. We have added this part of result in Figure 4-5 and the Result section (Figure 4H, line 216; Figure 5I, line 247). Corresponding references (ref 16, 17, 35) were added according to the reviewer’s comment.
- In Figure 3C I would correct the labeling of the graph.
We appreciate the reviewer’s constructive suggestion and comment. Since the results of metabolomic analysis indicated the normalized intensity of each metabolite, we corrected the labeling of Figure 3C as “normalized intensity of choline (â…¹107)” and “normalized intensity of phosphorylcholine (â…¹107)” (Figure 3C, line 189).
- Figure 3E: relative expression towards a control is missing in the labeling
We appreciate the reviewer’s constructive suggestion and comment. The results in Figure 3E showed the relative mRNA expression of CHKA from the transcriptome sequencing data of TJMUCH LUSC tissues. As the reviewer suggested, we revised the labeling of Figure 3E as “The relative mRNA expression of CHKA” and supplemented the description in detail in the revised manuscript (Figure 3E, line 189).
- Figure 3F-G: I would optimize the resolution of the images and especially for Figure G for which I have noticed a difference in the brightness of the pictures among single channels and the merged ones. Every picture has to be taken with the same intensity I would add also a quantification for that and a more detailed description in the method section.
We appreciate the reviewer’s constructive suggestion and comment. In the revised version, we optimized the resolution of the images from Figure 3F-G, as well as adjusted the pictures among single channels and the merged ones to the same brightness (Figure 3F-G, line 189). Furthermore, we supplemented a more detailed description in the section of Materials and Methods (line 379-line 392).
- Figure 4A: a proper labeling for mRNA expression is missing.
We appreciate the reviewer’s constructive suggestion and comment. In the revised version, we revised the labelling of Figure 4A as “The relative mRNA expression of CHKA”. All the RT-qPCR results shown for H520SH-ATP8B1 and H520SH-ATP8B1-si-CHKA were relative mRNA expression values compared to H520SH-NC while all the RT-qPCR results shown for SK-MES-1SH-ATP8B1 and SK-MES-1SH-ATP8B1-si-CHKA were relative mRNA expression values compared to SK-MES-1SH-NC (Figure 4A, line 216; Figure S2, line 601).
- Figure 4B GAPDH is missing in the labeling.
We appreciate the reviewer’s constructive suggestion and comment. We have supplemented the labeling of CHKA and GAPDH proteins in Figure 4B (Figure 4B, line 216).
- Figure 4G: labeling of the quantification should be consistent throughout the paper.
We appreciate the reviewer’s constructive suggestion and comment. We revised the Figure 4 and presented the results of H520 cell in main text. The results of another LUSC cell line SK-MES-1 were consistent which were listed in the supplementary for reference (Figure 4, line 216; Figure S2, line 601).
- Figure 6: the final figure needs some improvements, it lacks of labeling (COX is missing) and information both in the figure and in the figure legend and I would suggest to use a bigger font for ATP8B1 and explain the correlation with cardiolipin.
We appreciate the reviewer’s constructive suggestion and comment. We added the labeling in diagrams and explained all abbreviations in the figure and figure legend. We also supplemented the regulation of MAPK and PI3K/AKT signaling pathways by CHKA in diagrams to include all information. Furthermore, we have revised the font for ATP8B1 as the reviewer suggested (Figure S3, line 611-line 621).
- Check for typos and flow in the main text. Improve the method section and Figure legends as they lack informations.
In response, we thank the reviewer for the comment. We have checked for typos and flow and revised the main text in the new version. And we also supplemented the method section and Figure legends in detail.
Reviewer 2 Report
Abstract should be more concise and clear with clear conclusion of
ATP8B1 and REDOX
In the introduction and the Discussion elaborate more on cancer genomic networks including AKT gene regulation in cancer by REDOX
Fig 3,4,6, exclude all unnecessary figures
To the Fig 3 incorporate significant figures from supplements, no splemental figure is necessary, exclude
Fig 3 present gene expression, gene networks in correlation to the CHKA , and AKT and REDOX
Fig 3 show phosphorylated CHKA and phosphorylated AKT and their interaction
Author Response
Dear Reviewer,
Thank you for your letter stating your consideration of a revision of the manuscript entitled “ATP8B1 knockdown activated choline metabolism pathway and induced high-level intracellular REDOX homeostasis in lung squamous cell carcinoma”.
As indicated in your letter, we would like to resubmit our revised manuscript to Cancers for consideration of publication. My co-authors and I would like to thank you for their thoughtful comments and thorough review of our manuscript with the intent to strengthen our manuscript further. In acknowledgment of your comments, we are very pleased to recognize that you appreciate the potential significance and impact of our work. Therefore, in accordance with your comments we have added significant new data, described in detail below, and revised the manuscript to address your concerns. Below, we provide the following responses in a rebuttal to each comment.
- Abstract should be more concise and clear with clear conclusion of ATP8B1 and REDOX.
We appreciate the reviewer’s constructive suggestion and comment. We have revised the abstract accordingly (line 35-line 42).
- In the introduction and the Discussion elaborate more on cancer genomic networks including AKT gene regulation in cancer by REDOX.
We appreciate the reviewer’s constructive suggestion and comment. Since CHKA has been proposed to be an oncogene via activation of Ras and PI3K signaling, especially by increasing phosphorylated ERK and phosphorylated AKT, we detected the phosphorylation levels of ERK and AKT in H520SH-ATP8B1 cells and H520SH-ATP8B1-si-CHKA. We found that the levels of phosphorylated ERK (p-ERK) and phosphorylated AKT (p-AKT) were upregulated in H520SH-ATP8B1 cells, while downregulated after CHKA knockdown (Figure 4H, line 216). Furthermore, we added BSO (GSH inhibitor) to disrupt the cellular REDOX homeostasis in H520SH-ATP8B1 cells, and found that the levels of p-ERK and p-AKT were decreased significantly (Figure 5I, line 247). The results suggested that ATP8B1 knockdown promotes REDOX homeostasis through activation of CHKA, and then activates MAPK and PI3K/AKT pathways, thereby promoting tumor proliferation, invasion and migration. We have added this part of result in Figure 4-5 and the Result section (Figure 4H, line 216; Figure 5I, line 247). We have added the description of the relationship between AKT and REDOX in the Discussion section (line 308-line-314).
- Fig 3,4,6, exclude all unnecessary figures
We appreciate the reviewer’s constructive suggestion and comment. We revised the Figure 4 and present the results of H520 cell in main text. The results of another LUSC cell line SK-MES-1 were consistent which were listed in the supplementary for reference. We also listed the diagrams in supplementary rather than main figure (Figure 4, line 216; Figure S3, line 611).
- To the Fig 3 incorporate significant figures from supplements, no splemental figure is necessary, exclude
We appreciate the reviewer’s constructive suggestion and comment. We have revised the Figure 3 accordingly. In the revised version, we optimized the resolution of the images from Figure 3, as well as adjusted the pictures among single channels and the merged ones to the same brightness (Figure 3, line 189).
- Fig 3 present gene expression, gene networks in correlation to the CHKA, and AKT and REDOX
We appreciate the reviewer’s constructive suggestion and comment. The correlation among CHKA, and AKT and REDOX were added in Figure 4-5, as well as in the Result section (Figure 4H, line 216; Figure 5I, line 247). We found that the levels of p-ERK and p-AKT were upregulated in H520SH-ATP8B1 cells, while downregulated after CHKA knockdown (Figure 4H, line 216). Furthermore, we added BSO (GSH inhibitor) to disrupt the cellular REDOX homeostasis in H520SH-ATP8B1 cells, and found that the levels of p-ERK and p-AKT were also decreased significantly (Figure 5I, line 247). The results suggest that ATP8B1 knockdown promotes REDOX homeostasis through activation of CHKA, and then activates MAPK and PI3K/AKT pathways, thereby promoting tumor proliferation, invasion and migration.
- Fig 3 show phosphorylated CHKA and phosphorylated AKT and their interaction
We appreciate the reviewer’s constructive suggestion and comment. Aberrant choline phospholipid metabolism is a metabolic hallmark of cancer that has been implicated in tumorigenesis and progression. Among the multiple enzymes involved in the choline metabolism, CHKA is the first enzyme of the Kennedy pathway responsible for catalyzing the phosphorylation of free choline to form phosphocholine. As an important kinase, the activity of CHKA is usually characterized as the level of phosphocholine generation. Here we found the upregulation of CHKA protein expression in H520SH-ATP8B1 cells, as well as the level of phosphocholine, indicating CHKA was activated in H520SH-ATP8B1 cells. Further, since CHKA has been proposed to be an oncogene via increasing p-ERK and p-AKT, we also detected the phosphorylation levels of ERK and AKT. It showed p-ERK and p-AKT were elevated in H520SH-ATP8B1 cells and H520SH-ATP8B1-si-CHKA (Figure 4H, line 216). Corresponding references (ref 16, 17) were added according to the reviewer’s comment.
Round 2
Reviewer 2 Report
manuscript is corrected as recommended
now can be published